# Critical Influence of Water on the Polymorphism of 1,3-Dimethylurea and Other Heterogeneous Equilibria

**DOI:** 10.3390/molecules28207061

**Published:** 2023-10-12

**Authors:** Grace Baaklini, Manon Schindler, Lina Yuan, Clément De Saint Jores, Morgane Sanselme, Nicolas Couvrat, Simon Clevers, Philippe Négrier, Denise Mondieig, Valérie Dupray, Yohann Cartigny, Gabin Gbabode, Gerard Coquerel

**Affiliations:** 1Laboratoire Sciences et Méthodes Séparatives UR3233, Université Rouen Normandie, Normandie Université, F-76000 Rouen, France; grace.baaklini@dsm-firmenich.com (G.B.); manon.schindler@servier.com (M.S.); lina.yuan@novartis.com (L.Y.); clement.de-saint-jores@univ-orleans.fr (C.D.S.J.); morgane.sanselme@univ-rouen.fr (M.S.); nicolas.couvrat@univ-rouen.fr (N.C.); simon.clevers@gmail.com (S.C.); valerie.dupray@univ-rouen.fr (V.D.); yohann.cartigny@univ-rouen.fr (Y.C.); 2LOMA, UMR 5798, Université Bordeaux, 351 Cours de la Libération, F-33400 Talence, France; philippe.negrier@u-bordeaux.fr (P.N.); denise.mondieig@u-bordeaux.fr (D.M.)

**Keywords:** phase diagrams, polymorphism, hydrate, phase transitions, crystal structure determination

## Abstract

It is shown that the presence of hundreds of ppm of water in 1,3-dimethylurea (DMU) powder led to the large depression of the transition temperature between the two enantiotropically related polymorphic forms of DMU (Form II → Form I) from 58 °C to 25 °C, thus explaining the reported discrepancies on this temperature of transition. Importantly, this case study shows that thermodynamics (through the construction of the DMU–water temperature-composition phase diagram) rather than kinetics is responsible for this significant temperature drop. Furthermore, this work also highlights the existence of a monohydrate of DMU that has never been reported before with a non-congruent fusion at 8 °C. Interestingly, its crystal structure, determined from X-ray powder diffraction data at sub-ambient temperature, consists of a DMU–water hydrogen bonded network totally excluding homo-molecular hydrogen bonds (whereas present in forms I and II of DMU).

## 1. Introduction

Organic and pharmaceutical compounds can interact with water during production or upon storage under a wet atmosphere [1,2,3,4,5,6]. Because of its small size and multidirectional hydrogen bonding capability, a water molecule can easily be incorporated into a crystalline structure, resulting in hydrated forms [7,8,9,10,11,12] or in the alteration of the quality and the physical properties of the powder [13,14,15,16]. Further, the energy landscape between polymorphs can be modified with the addition of extra components via the formation of partial solid solutions [17].

The hardly predictable interactions in the solid state between two compounds can be better monitored and understood by exploring heterogeneous equilibria at different temperatures and/or pressures.

In this regard, this article deals with a case study of water and 1,3-dimethylurea (hereafter DMU) used as an intermediate compound for the synthesis of caffeine, chemicals, textile aids and herbicides [18,19,20]. DMU has also been used as an excipient capable of blocking at room temperature the (metastable) high-temperature form of the active pharmaceutical ingredient pyrazinamide for an extremely long period of time [21].

A literature review on the DMU molecule (Figure 1) indicates that it crystallizes in two enantiotropically related polymorphic forms [22], named Form I and Form II. Form I is the stable polymorph at high temperature and it crystallizes in the rather uncommon *F*dd2 space group (refcodes: NIJHUJ01 and 03—occurrence 0.3% in the CSD) while Form II is the stable form at low temperature, which crystallizes in the *P*2_1_2_1_2 space group (refcodes: NIJHUJ02 and 04) with Z’= 1/2; thus, the DMU molecule is in a special position with superimposition between the molecular symmetry and the crystallographic 2-fold axis. Nevertheless, the solid–solid transition temperature between both forms remained unclear. Indeed, according to [22], it was detected at 42 °C by differential scanning calorimetry (DSC) analysis, but was also observed at −20 °C. Furthermore, the authors found that Form I is the most stable form at room temperature from precise slurry experiments performed at room temperature in many solvents. This finding is not consistent with the observation of a solid–solid transition at 42 °C and thus we decided to further investigate the polymorphic behavior of DMU.

According to the Gibbs phase rule, a fixed solid–solid transition temperature is mandatory in a unary system if pressure is fixed. Thus, different temperatures for the same polymorphic transition can come from: (i) experimental errors, (ii) problems related to the kinetics [23,24], and (iii) thermodynamics [25]. If we consider that the two first reasons do not come into play, then the third one needs careful examination. Indeed, from a thermodynamic point of view, this temperature variation can be associated with the presence of an impurity: thus, from the unary system, it does a swap to a binary system.

Figure 2 shows the three possible situations that can be encountered with three different additives as a function of its concentration (in arbitrary unit). Impurity B (Figure 2a) is the case where no solid solution exists. This impurity cannot enter in the crystal lattices of Form I and Form II without a high energetic penalty; this impurity is neutral for the polymorphic transition (blue horizontal line in Figure 2a). By contrast, impurity C (Figure 2b) can enter in both crystal lattices, Form I and Form II, but with more affinity for the high-temperature Form I. The temperature of transition is depressed to a metatectic invariant (green horizontal line in Figure 2b): T_meta_.
At T_meta_: solid solution I (X_1_) ⇔ solid solution II (X_2_) + liquid (X_L_)(1)
where X_1_, X_2_ and X_L_ are the fixed compositions of the binary mixtures of the corresponding phases.

From left to right in Equation (1) (solid solution I transforming into a mixture of solid solution II and liquid—a doubly saturated solution), the enthalpy of transition ΔH is negative (crystallization of solid solution II upon cooling) and as ΔG = 0 (this is an equilibrium), one can thus infer that ΔS < 0 (ΔS = S_final_ − S_initial_). This appears unusual as the initial solid leads to another solid plus a liquid (high internal disorder). The sign of a variation in entropy from left to right is indeed negative because solid solution I is extensively disordered (S(ssI) > S(ssII)) and most of the time X_2_ is much closer to X_1_ than X_L_. At the temperature of the invariant—T_meta_—when removing heat from the system, the solid solution ssI of composition X_1_ just sweats a small amount of doubly saturated solution (its contribution to entropy is thus very small) and solid solution ssII of composition X_2_.

Impurity D (Figure 2c) leads to the reverse situation to that presented with impurity C. Molecules D can also enter in the crystal lattices of Form I and Form II but have a greater affinity for the low-temperature form (polymorph II) rather than the high-temperature form (polymorph I). The upper temperature of existence of polymorph II is the temperature of the peritectic invariant (red horizontal line in Figure 2c): T_P_.
At T_P_: solid solution II (X_2_) ⇔ solid solution I (X_1_) + liquid (X_L_)(2)

Thus, the existence of solid solutions induces variations in the temperature of polymorphic transitions and the existence of a domain in temperature and composition where they can coexist in equilibrium. Conversely, if a temperature of polymorphic transition fluctuates without a kinetic reason or any other bias, it can be concluded that an impurity can enter in at least one of the two polymorphs.

DMU has a strong affinity with water [26] and discrepancies on the temperature of transition between forms I and II are reported [22]. It is possible that water influences the temperature of the solid–solid transition (in terms used in Figure 2, DMU will act as constituent A and water as the “impurity”) as in the ammonium nitrate–water system at ppm levels [25]. Furthermore, repeated heating–cooling cycles by DSC on the same sample of DMU could show the reproducibility of the polymorphic transition (see Appendix A), thus disregarding any kinetic origin of the observed changes on the polymorphic transition temperature. For these reasons, it is worthwhile to reinvestigate the solid–solid transition of DMU in particular with thorough control of the amount of water in the samples to account for the role of water on the polymorphic behavior of DMU. In this prospect, the entire DMU–water binary phase diagram will be explored and built with the support of several techniques such as DSC, ex situ and in situ X-ray diffraction and second harmonic generation (SHG). In the present manuscript, the composition of the DMU–water mixtures will be always expressed in molar fraction (in %) of water unless otherwise stated.

## 2. Results

### 2.1. Influence of Water on DMU Form II—Form I Solid–Solid Transition Temperature

#### 2.1.1. Re-Investigation of the Temperature of Transition

As previously mentioned, the Form II to Form I polymorphic transition is reported in the literature at temperatures ranging between −20 °C and 42 °C. However, when DMU powder (Form II) is stored during 6 months in a desiccator under dry atmosphere (P_2_O_5_, 0% relative humidity—RH), the II to I transition temperature is recorded at ca. 58 °C from DSC measurements (Figure 3a; the DSC crucible has been filled in the desiccator then directly sealed). This phenomenon is followed by an endothermic peak with an onset temperature at 106.1 °C referring to the melting of Form I.

Form II of DMU was also analyzed by temperature-resolved X-ray powder diffraction (TR-XRPD) and the solid–solid transition temperature was found between 30 °C and 40 °C (Figure 3b).

Thus, DSC and TR-XRPD analyses performed on Form II display quite different values for the same solid–solid transition temperature. To explain this discrepancy, it is interesting to compare the atmosphere surrounding the sample during each analysis. In fact, all along the DSC analysis, the relative humidity is controlled by the dry nitrogen flow and supposed to be close to 0% and thus the sample which was already stored at 0% RH for a long period of time could not uptake water during the measurement. By contrast, the TR-XRPD analyses were performed in a close chamber sealed under ambient atmosphere (RH of 50–60% at 20 °C). Consequently, the powder placed on the sample holder could easily absorb water and the solid–solid transition temperature decreased following a metatectic equilibrium (see Figure 2b) associated with the solid–solid transition of DMU in the presence of water even if present in a minute quantity. The exploration of the DMU–water phase diagram is then necessary to prove or reject this hypothesis.

#### 2.1.2. In-situX^®^ Analyses of DMU Samples Crystallized in the Presence of Water

A first series of In-situX^®^ diffraction analyses (see Section 4.2.1. for more details on the experimental setup) was carried out on DMU Form II in n-heptane already dried by using a molecular sieve. We predict that n-heptane acts here as a diluent as DMU is poorly soluble in n-heptane. The amount of water in the reactor was estimated at 40 ppm by Karl Fischer titration, corresponding to a molar composition of 0.15% in water (see Appendix A). The system was cooled to 12 °C in order to stay in the domain of stability of Form II. The XRPD analyses show that Form II crystallizes solely in the temperature range 12–28 °C. Peaks referring to Form I appear above 28 °C and increase in intensity upon heating. A domain of coexistence of the solid solution Form I and the solid solution Form II (noted ssI + ssII) is thus detected between 28 °C and 42 °C (Figure 4a). Above 42 °C, Form I crystallizes solely. In reference to the DSC analysis, the solid–solid transition of pure DMU Form II → Form I was found at 58 °C. Thus, when the system contains 0.15% molar in water, the solid–solid transition decreases from 58 °C to 28 °C. We made sure that during the In-situX^®^ monitoring at different temperatures, the coexistence of the two polymorphs was not due to a slow kinetics of conversion but due to the genuine thermodynamic equilibrium (see experimental section).

Figure 4b illustrates a second series of In-situX^®^ diffraction analyses performed on DMU in a suspension with n-heptane saturated in water prepared by mixing and stirring n-heptane with water during 24 h. The amount of water was found at 98 ppm of water by Karl Fischer titration, corresponding to 0.39% molar in water (see Appendix A). At this composition, the solid–solid transition from Form II to Form I is detected at 24 °C. The domain of coexistence ssI + ssII extends between 24 °C and 36 °C.

A last series of In-situX^®^ diffraction analyses was performed by adding 200 microliters of water to the reactor (molar composition 12%). Thus, the solid–solid transition from Form II to Form I was found at 26 °C. No domain of coexistence of ssI + ssII could be detected below or above that temperature: only peaks of structurally pure Form I and peaks of structurally pure Form II were detected (Figure 4c).

In light of these results, it can be deduced that the addition of water results in the depression of the solid–solid transition following a metatectic equilibrium. When the molar fraction in water in the reactor was increased from 0.15% to 0.39%, the lower limit of the coexistence domain of the two forms decreased from 28 °C to 24 °C. At the composition 12% molar in water, the transition temperature (26 °C) occurs without any noticeable domain of coexistence of ssI + ssII. Hence, the metatectic invariant is likely to be at 25 ± 1 °C considering the temperature of transition of the two last mixtures. The first mixture serves to locate below (but close to) the lower limit of the metatectic invariant (that is below composition x_2_ in Figure 2b) then undergoing the ssII–ssI transition by crossing the ssII–ssI solvus inferior line.

### 2.2. Identification of the DMU Monohydrate and Resolution of Its Crystal Structure

The peculiar affinity of DMU for water is further evidenced by the stabilization of a DMU monohydrate, which is, to the best of our knowledge, reported in the present work for the first time. When X-ray diffraction is performed on DMU–water mixtures of molar composition in the range 50–60% in water cooled to ca. −25 °C, DMU Form II is obtained. However, if the sample is held at this temperature for a long period of time (typically several weeks), a singular X-ray diffraction pattern can be observed that is different from that of the polymorphs of the components (see Appendix A). Upon heating, it vanishes and is replaced by that of DMU Form II at a temperature which is between 0 °C and 11 °C (see Appendix A). These results suggest the existence of a DMU monohydrate with a non-congruent melting, e.g., characterized by a peritectic invariant below 11 °C. To confirm this hypothesis, crystal structure resolution of this new solid phase has been attempted from X-ray powder diffraction data as its crystallization is difficult and did not afford the growth of large enough single crystals. Nevertheless, we found that performing heating—cooling cycles between −25 °C and −10 °C considerably fastened the crystallization of the hydrate which thereby could occur within several hours. Furthermore, this relatively smooth crystallization leads to a significant reduction in preferential orientations of crystallites which are otherwise quite strong on direct crystallization from the melt. High-quality XRPD patterns could then be measured at −20 °C for a DMU–water mixed sample of molar composition 51% in water from which the crystal structure of the monohydrate could be solved with a fairly good agreement factor (Rwp = 0.048) as shown in Figure 5. Noticeably, two reflections at 16.9° and 25.5° in 2θ (spotted by black arrows in Figure 5) that could not be indexed using the unit cell parameters of neither of the two known polymorphs of anhydrous DMU nor those of ice were revealed during the heating–cooling cycles. The intensity of both reflections decreased continuously with the number of cycles but still remain non-negligible after several cycles (see Appendix A). This indicates that those two reflections do not belong to the XRPD pattern of the DMU monohydrate but rather refer to a possible new metastable polymorph of either the DMU monohydrate or anhydrous DMU. Accordingly, both reflections were not considered for the crystal structure determination of the DMU monohydrate.

The crystal structure of the DMU monohydrate consists of a monoclinic *C*2/c lattice containing 16 molecules of DMU and 16 molecules of water (thus confirming the 1:1 stoichiometry of the hydrate) in the unit cell as shown in Table 1. Hence, the asymmetric unit is composed of two symmetry independent DMU molecules and two symmetry independent water molecules (Z′ = 2). The unit cell volume is approximately two and a half times larger than that of DMU Form I (Z = 8) and approximately ten times larger than that of DMU Form II (Z = 2). Yet, the density is comparable to that of forms I and II (1.099 g/cm^3^ to be compared to 1.120 and 1.151 g/cm^3^, see Table 1). This suggests a tight packing arrangement of the molecules in the monohydrate crystal structure as for the anhydrous DMU polymorphs.

The most striking feature of the crystal packing of the DMU monohydrate is that no DMU—DMU nor water—water hydrogen bond interaction is present, each DMU molecule being connected to four surrounding water molecules through hydrogen bond interactions (Figure 6b). However, the crystal structures of forms I and II of anhydrous DMU exclusively lie on the packing of infinite molecular chains made of DMU molecules connected with each other through bifurcated N-H∙∙∙O bonds (Figure 6a). Their crystal structures mainly differ in the packing of the molecular chains which are all oriented in the same direction for Form I (i.e., polar crystal) while they are alternatively in opposite direction for Form II. The II to I phase transition might thus involve a reversible re-orientation of the chains upon heating (or cooling for the reverse transition). Therefore, a non-negligible amount of energy should be required to “destroy” the homo-molecular chains present in anhydrous DMU Form II and build the hetero-molecular chains present in the DMU monohydrate when temperature decreases below the peritectic temperature. This might explain the observed slow kinetics of the crystallization of the DMU monohydrate.

Table 2 gathers the distances and angles of DMU–water hydrogen bonds for the DMU monohydrate together with those of DMU—DMU hydrogen bonds for forms I and II of anhydrous DMU. It appears that the N-H∙∙∙O angles and the O∙∙∙N distances of the hydrogen bonds are quite similar when comparing the DMU monohydrate and the anhydrous forms, suggesting hydrogen bonds of comparable strength (the values of angles for the monohydrate have to be taken with care as H atoms are less precisely located). Concerning the O–H∙∙∙O hydrogen bonds, the angles are similar to the N-H∙∙∙O angles on the average and the O∙∙∙O distances are a little shorter than the O∙∙∙N distances. This latter result is consistent with the higher electronegativity of oxygen atom compared to nitrogen atom. More details on the crystal packing of the DMU monohydrate can be found in Appendix A.

### 2.3. Exploring the Complete DMU–Water Phase Diagram

#### 2.3.1. DSC Analyses of Closed Crucibles Enriched with Water

The DMU–water binary phase diagram was completed with the DSC data obtained for the molar compositions ranging from 20% to 95% in water. The first DSC analyses carried out without annealing (Figure 7a) revealed a complex behavior with several overlapping phenomena:An endotherm located at circa −36 °C directly followed by an exothermic phenomenon (recrystallization, emphasized by the blue rectangle in Figure 7a). Hence, this endotherm would likely belong to a metastable equilibrium.An endotherm at −20 °C can be assigned to the presence of a eutectic equilibrium between the monohydrate and ice.A third endotherm located at circa 8 °C, the maximum heat exchange of which is observed for a molar composition in water of 50% (see Appendix A) that can be fairly attributed to the non-congruent fusion of the monohydrate.A fourth endotherm at approximately 25 °C which corresponds to the metatectic invariant also previously revealed by In-situX^®^ measurements (see Section 2.1.2).A fifth endotherm located at different temperatures corresponding to the end of fusion of the solid solution I (e.g., liquidus line).

The above assignment of the different thermal events has been achieved by combining the collected temperatures for all mixtures investigated and the knowledge on the solid forms exhibited by the different mixtures, gained from X-ray diffraction experiments.

In a second set of experiments, and to get closer to equilibrium, an annealing at −10 °C for two hours was performed for each composition in DSC before starting the heating procedure. Examples of DSC heating curves obtained before and after annealing representing the metastable and stable equilibria are shown in Figure 7.

After annealing, only the peritectic invariant (8.3 °C), the solid–solid transition at circa 25 °C (metatectic invariant) and the liquidus (58 °C) were recorded for the 40% molar composition in water. By contrast, for the 53 and 58% molar compositions in water, stable eutectic invariants are present at circa −20 °C, followed by the peritetic invariant, the solid–solid transition at circa 25 °C (metatectic invariant) and finally the liquidus (Figure 7b).

The DSC data are represented as blue dots on the phase diagram illustrated in Section 2.3.3.

#### 2.3.2. Refractometry Measurements

For most binary mixtures, the refractive index changes linearly over a wide range of concentrations (expressed in mass fraction, see Appendix A), thus the solubility of a compound in a given solvent at a given temperature can be determined using refractometry. Accordingly, the solubility curve of DMU in water between −15 °C and 16 °C has been constructed by the refractometry method and it was represented by plotting the temperatures as a function of the molar fraction (in %) of water (Figure 8). It is likely to correspond to the metastable eutectic invariant between DMU ssII and water. A sharp decrease in the temperature (from 16 °C to −15 °C) is noticed on a small range of molar fraction of water (from 62% to 72%). These data are reported on the binary phase diagram as black dots and also gathered in Appendix A.

#### 2.3.3. Refinement of the Metastable Eutectic Composition of DMU–Water Binary System by Temperature-Resolved Second Harmonic Generation (TR-SHG)

Second harmonic generation (SHG) is a fast and sensitive non-linear optical technique used in particular for the precise determination of eutectic compositions [27] as well as the investigation of solid–solid transitions [28,29,30]. This technique requires at least one non-centrosymmetric crystal phase. Thus, TR-SHG technique can be used in this study since DMU Form II crystallizes in the non-centrosymmetric space group *P*2_1_2_1_2. The refinement of the eutectic composition will be then determined by TR-SHG.

Different compositions of DMU–water mixtures were prepared and analyzed by TR-SHG. Figure 9a illustrates a TR-SHG analysis performed on a mixed sample of composition 70% in molar fraction of water. The curve shows that in the region colored in green, the signal remains stable between −50 °C and −37.5 °C. In this region, a mixture of ice and DMU (ssII) crystals exists, in which DMU (ssII) crystals are responsible for the high SHG signal since ice crystallizes in the centrosymmetric space group *P*6_3_/mmc. The SHG signal sharply decreases at −37.5 °C which corresponds to the metastable eutectic temperature. To be noticed, the latter temperature is fairly close to that determined from DSC experiments (−36 °C, see Section 2.3.1). In the region colored in yellow, DMU crystals disappear progressively; therefore, the SHG signal decreases progressively and totally vanishes at −17.5 °C, which corresponds to the liquidus temperature. Therefore, the composition of 70% molar in water is a hypoeutectic composition since both the metastable eutectic temperature and the liquidus temperature can be determined (transition from the two-phase domain DMU Form II + Ice to the two-phase domain DMU Form II + Liquid at the eutectic temperature). The TR-SHG results for the composition of 70.5% in molar fraction of water (Figure 9b) show a plateau from −50 °C to −37.5 °C (region colored in green). In this domain of temperature, a mixture of DMU ssII and ice exists, as for the previous experiment. Once the metastable eutectic temperature (−37.5 °C) is reached, the TR-SHG signal decreases sharply, thus indicating that no liquidus temperature is observed above −37.5 °C (region colored in yellow). It means that the composition of 70.5% in molar fraction of water corresponds to a metastable hypereutectic composition (transition from the two-phase domain: DMU Form II + Ice to the two-phase domain: Ice + Liquid at the eutectic temperature).

Thus, the metastable eutectic composition (noted E’ in Figure 10) is located between the molar compositions 70% and 70.5% in water. The little broad peak observed at approximately −30 °C after the sharp intensity decrease in Figure 9a might be due to a change in the number and size of ssII crystals upon crossing the eutectic invariant [31,32].

The SHG data are represented as red dots on the phase diagram presented in Figure 10 and also reported in Appendix A.

#### 2.3.4. Plotting the DMU–Water Binary Phase Diagram

Combining results from DSC, In-situX^®^ diffraction, TR-XRPD, TR-SHG and refractometry analyses, the binary phase diagram between DMU and water is built and represented in molar fraction of water in Figure 10.

In the inset, the two domains of solid solution are magnified in water composition for clarity. It can be deduced that the system exhibits a metatectic equilibrium with low concentration of water (hundreds of ppm—mass fraction) which sharply decreases the polymorphic transition of DMU from 58 °C to 25 °C.

A peritectic invariant is revealed at 8 °C above which Form II crystallizes as a solid solution containing a minor proportion of water. The peritectic point P has been determined at approximately 50% in molar fraction of water from DSC measurements (Tammann graph, see Appendix A). This value is consistent with the stabilization of a DMU monohydrate at low-temperature (below 8 °C) whose stoichiometry has been confirmed by the resolution of its crystal structure (see Section 2.2).

The stable eutectic composition E was found at 80% molar in water by extrapolation of the liquidus line of ice.

The eutectic composition E’ referring to the metastable equilibrium between ice and ssII (represented in blue dashed lines) was localized between 70% and 70.5% in molar fraction of water as deduced from TR-SHG results.

It should be noted that a metastable equilibrium should also exist between ssI and ice with an eutectic invariant likely below −37 °C but this one has never been observed in this work.

Characteristic information (temperatures, phases involved, …) on the various invariants detected by DSC are summarized in Table 3.

This binary phase diagram unambiguously shows that the temperature of the solid–solid transition between forms II (as ssII) and I (as ssI) of DMU is crucially dependent on the amount of water. Therefore, the thermodynamics of heterogeneous equilibria rules the temperature of this solid–solid transition.

## 3. Discussion

The In-situX^®^ analyses of DMU carried out in n-heptane with different amounts of water confirmed the existence of a metatectic invariant at 25 ± 1 °C with the presence of 0.15% of water in molar fraction. Hence, water at the hundreds of ppm level decreases the polymorphic transition temperature of DMU from 58 °C to 25 °C, leading to discrepancies in the temperature of transition between Form I and Form II reported in particular in [22], which are thus likely to result from a very small variation in the water content of DMU samples. In [22], the authors stated that “the true thermodynamic transition temperature lies somewhere between 253 K and room temperature”. Their conclusion was based on systematic solvent-mediated conversion experiments carried out with organic solvents that are all prone to contain different amounts of water (acetone, ethanol, 1-propanol, 2-propanol, THF, dioxane, DMSO -dimethyl sulfoxide-, chloroform, carbon tetrachloride, dichloromethane, tert butyl methyl ether, diethyl ether, ethyl acetate, 2-butanone) that were not estimated in their manuscript. However, one can notice that their suggested lowest limit of the transition temperature (253 K = −20 °C) corresponds to the stable eutectic between DMU and water which means that their systems definitely contained an appreciable amount of water while no drying operations of the samples were reported by the authors. Consequently, the metatectic equilibrium between water and DMU much likely induced the depression of the transition temperature, thus preventing the authors from finding the respective stability domains of the two DMU polymorphs in dry conditions. Indeed, with even a small amount of water in the environment, DMU Form I (solid solution actually) then becomes more stable at 25 °C and Form II is thus hardly observed at room temperature.

With the support of DSC, TR-SHG data, solubility measurements retrieved from different compositions of DMU–water, it was possible in this study to detect several stable and metastable invariants including a peritectic at 8 °C with a maximum heat exchange for the equimolar composition. This invariant was assigned to the non-congruent fusion of a stoichiometric monohydrate of DMU crystallizing in the space group *C*2/c. The crystal packing of the DMU monohydrate did not show any hydrogen bond interaction between DMU molecules as those present in the polymorphs I and II. Thus, the observed slow crystallization of the DMU monohydrate in water from Form II can be justified by this particular crystal packing involving a complete reshaping of the H-bond network from Form II (actually the solid solution) to the monohydrate.

As a final conclusion, this case study is a clear illustration of the importance of investigating heterogeneous equilibria to fully understand the polymorphic behavior of crystalline materials. Notably, careful inspection of the binary phase diagram should be undertaken when a variation in the temperature of transition between two polymorphs (with enantiotropic relationship) is observed for solutes having a high solubility in water and/or as a result of solution maturation studies performed at different temperatures with various solvents.

## 4. Materials and Methods

### 4.1. Materials

DMU was acquired from Merck Schuchardt OHG and it was recrystallized from ethanol upon two cycles. The purity of the recrystallized DMU is estimated at 99.9% molar by using NETSCH purity software. The two main impurities determined by HPLC are N-methylurea and 1,1-dimethylurea (see Appendix A). Before use, the recrystallized DMU was stored several weeks in a desiccator containing P_2_O_5_ for drying.

Form II of DMU was obtained by preparing a suspension of DMU in n-heptane. The suspension was stirred for 24 h in a cold room at 4 °C, then filtered, dried and stored at the same temperature.

### 4.2. Methods

#### 4.2.1. In-situX^®^ Analyses and Compositions Preparation

In-situX^®^ diffraction data were collected using the prototype diffractometer [33], a schematic description of which is given in Appendix A. In-situX^®^ is equipped with an original goniometer with a reverse geometry (−θ/−θ). Its association with a dedicated reactor, with its bottom part being transparent to X-ray, allows us to perform in situ X-ray diffraction analyses of the solid in suspension. The incident beam is produced by an X-ray tube with a copper anode while diffracted X-rays are collected by a linear Lynxeye^TM^ (Bruker, Germany) detector.

In-situX^®^ analyses were performed on 7 g of DMU Form II in suspension with 80 mL n-heptane with different amounts of water (0.15%, 0.39%, 12% in molar fraction). Measurements were carried out in the [14–30°] 2θ range, with a step size of 0.04° and a step time of 0.5 s/step (measurement time of approximately 3 min). X-ray diffraction measurements were performed every 15 min at several temperatures in the range 12–46 °C. The temperature was increased in a step of 1 or 2 °C from the initial temperature (12 °C or 18 °C) at a heating rate of 1 °C/min and was kept constant during one hour at each step. Thereby, at least 4 X-ray diffraction patterns were acquired per temperature at increasing equilibration time, thus allowing us to ensure that thermodynamic equilibrium was effectively reached at each measurement temperature. The uncertainty on the boundary temperatures of the different phase domains revealed by In-situX^®^ analyses is set as ±2 °C.

#### 4.2.2. DSC Measurements and Compositions Preparation

DSC analyses were performed using a DSC 214 Polyma (Netzsch). Samples were weighed in 25 µL aluminum pans and submitted to a heating rate of 5 °C/min under a dry nitrogen atmosphere. Data treatment was performed by using Netzsch Proteus^®^ software v6.1.

As far as the construction of binary phase diagrams is concerned, the onset temperature of the observed peaks was considered for the melting and solid–solid transition of the pure compounds and for solvus inferior, solidus and invariant transitions of binary mixtures. For the liquidus, the peak value was considered.

The various DMU–water mixed samples were analyzed by DSC in the temperature range [−50–80 °C] with a heating rate of 5 °C/min.

Crucibles containing recrystallized DMU were placed in a desiccator at 20 °C with a saturated salt solution of sodium chloride in order to control the relative humidity (75% at 20 °C) due to its deliquescent character [26]. DMU samples were then enriched in water. The percentage of water absorbed in each powder sample was determined by weighing accurately the crucible before and after storage in the desiccator. Furthermore, after this conditioning, crucibles were hermetically sealed with lids in order to keep the amount of water constant all along DSC analyses. The latter were started immediately after sample preparation.

#### 4.2.3. Temperature-Resolved Second Harmonic Generation (TR-SHG) and Mixtures Preparation

Second harmonic generation is an optical process that can occur upon exposure of a crystalline solid to a high power laser beam by which the sample re-emits a wave of half the wavelength (double frequency) of the incoming wave. The latter process is possible only if the sample crystallizes in a non-centrosymmetric crystal structure. This is true for both DMU forms II (space group *P*2_1_2_1_2) and I (space group Fdd2). Two DMU–water mixtures were analyzed by TR-SHG. Temperature was monitored using a Linkam hot stage which was systematically purged with N_2_ before each experiment, so that the analyzed sample was in an inert atmosphere during the measurement. The solutions were first cooled to −50 °C at a cooling rate of 10 °C/min, then kept at −50 °C for two hours and finally heated at 5 °C/min. TR-SHG measurements were performed every minute during the heating process with a measurement time of 3 s each.

#### 4.2.4. Refractometry Analyses

A calibration curve was first plotted by measuring the refractive index of eight DMU aqueous solutions of known concentrations (ranging from 50 to 100% by weight in water) at room temperature. Furthermore, a suspension of DMU in water was prepared and stirred for 24 h at a given temperature ranging from −15 °C to 16 °C. The refractive index of the saturated solutions at a given temperature was measured by refractometry. Thus, the molar composition was determined with reference to the calibration curve, thus affording the construction of the solubility lines of DMU in water in the above-mentioned range of compositions.

#### 4.2.5. Temperature-Resolved X-ray Powder Diffraction (TR-XRPD)

TR-XRPD diffraction data were collected using a D5005 diffractometer equipped with a TTK 450 heating stage from Anton Paar. X-rays are produced from a copper source monochromatized using a Kβ Ni filter. Typical X-ray diffraction patterns were measured in the 10–30° 2θ range, using a scanning step of 0.04° and a counting time of 4 s/step. Low temperatures (below 20 °C to −25 °C) were reached with the support of a Lauda RP890 cryostat (Lauda-Königshofen, Germany). Samples were heated at a rate of 2 °C/min between the measurements, and maintained 30 min at every measuring temperature.

#### 4.2.6. Crystal Structure Determination from X-ray Powder Diffraction Data

X-ray powder diffraction patterns were measured using an INEL diffractometer comprising a horizontally mounted cylindrical position-sensitive gas-filled (Argon + C_2_H_6_) CPS 120 detector and operating with a (curved) quartz monochromatized X-ray beam ($CuK_{\alpha_{1}}$ radation, λ = 1.54056 Å) in a Debye Scherrer type transmission geometry. Diffracted rays are collected simultaneously on 120° by the 4096 channels of the detector, the conversion in 2θ positions (degrees) of which is performed through a calibration procedure using cubic Na_2_Ca_2_Al_2_F_4_ (high angle calibration) [34] mixed with silver behenate (low angle calibration) [35]. Powder samples are introduced into 1 mm diameter glass capillaries that rotates around their axis all along the measurement to reduce preferential orientations of crystallites. Sample temperature is controlled accurately (±0.1 °C) by a liquid nitrogen fed 600 series Cryostream cooler from Oxford Cryosystems.

Unit cell determination, crystal structure resolution and refinement were achieved using procedures [36,37,38] included in the Reflex Plus module of Materials Studio software [39]. In particular, a rigid body-based direct space method was used to solve the crystal structure from the experimental X-ray powder data. By this method, the molecules of the asymmetric unit with fixed geometry (for DMU molecule bond distances and angles were extracted from the known crystal structures of forms I and II [22]) were randomly displaced, together with their symmetry homologues, in the unit cell using a Monte Carlo type (simulated annealing) routine [38], which generates thousands of trial structures by varying the degrees of freedom of the molecules (3 translations and 3 rotations). After several cycles, the crystal structure affording the best agreement between calculated and measured diffracted intensities was selected. This solution was further refined by Rietveld refinement [40], which included refinement of peak profile parameters, unit cell parameters, degrees of freedom of the molecules, a global isotropic temperature factor and parameters accounting for the effects of preferential orientations of crystallites [41].

## 5. Conclusions

This case study emphasizes the huge impact of water on the polymorphic behavior of 1,3-dimethylurea, a low molecular weight organic compound mainly used as an intermediate in the synthesis of caffeine. It is shown that the temperature of the solid solid transition between its polymorphic forms (named forms I and II) drops of more than 30 degrees in the presence of ppm amounts of water. We prove that thermodynamics rather than kinetics is responsible of this significant temperature change which is characterized by a metatectic temperature invariant indicating the formation of solid solutions of forms I and II with the addition of water. The influence of water on the polymorphic behavior of 1,3-dimethylurea is further revealed by the discover of a monohydrate stable below around 8 °C, the crystal structure of which have been determined in this study from X-ray powder diffraction data. We hope this work will aid the community of solid state scientists in the difficult understanding of polymorphism and polymorphic transitions which have a significant impact in many research fields.

## Figures and Tables

**Figure 1 molecules-28-07061-f001:**
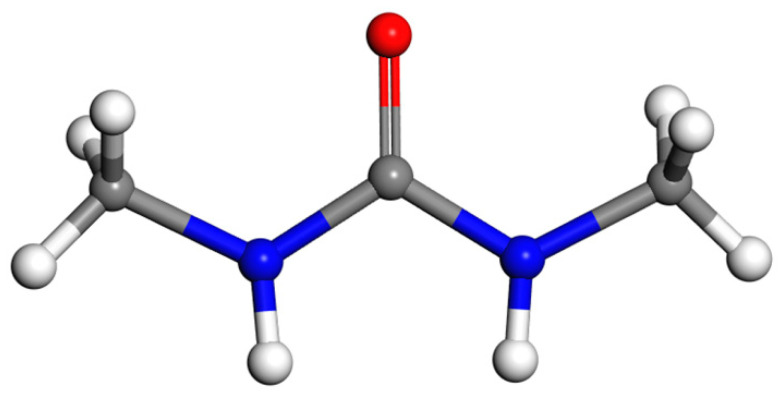
Molecular structure of 1,3-dimethylurea (DMU). Color code: carbon, nitrogen, oxygen and hydrogen atoms are in grey, blue, red and white, respectively.

**Figure 2 molecules-28-07061-f002:**
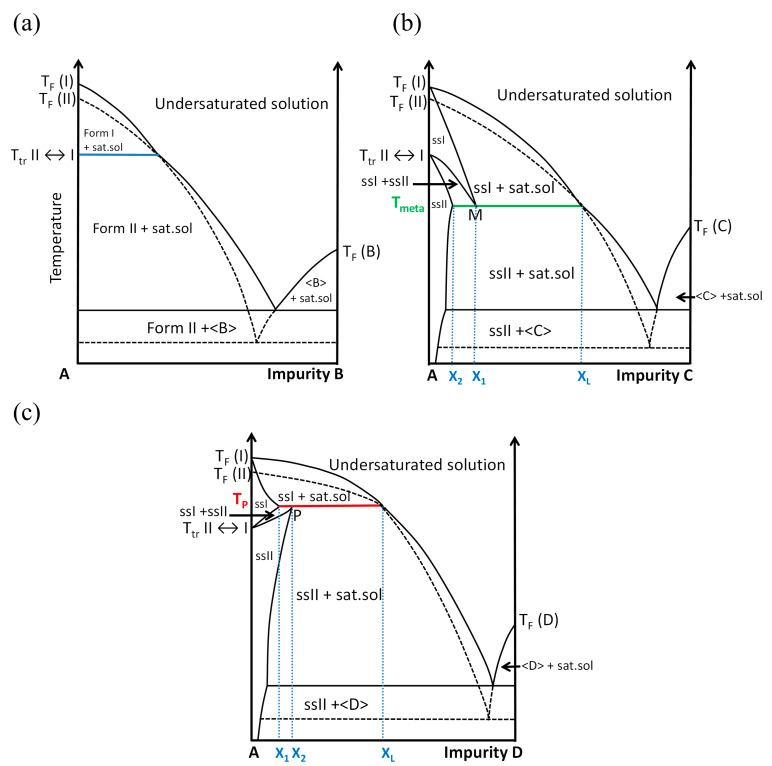
Impact of the presence of an impurity on the temperature of polymorphic transition. (**a**) B is a neutral impurity, (**b**) C is an impurity which drops the temperature of transition because it has more affinity with the high-temperature Form I than with the low-temperature Form II, and (**c**) D is an impurity which increases the temperature of transition because it has more affinity with the low-temperature Form II than with the high-temperature Form I. Glossary: ss = solid solution, sat. sol. = saturated solution, T_F_ = melting temperature, and T_tr_ = temperature of the reversible solid–solid transition. Dashed lines represent metastable equilibria.

**Figure 3 molecules-28-07061-f003:**
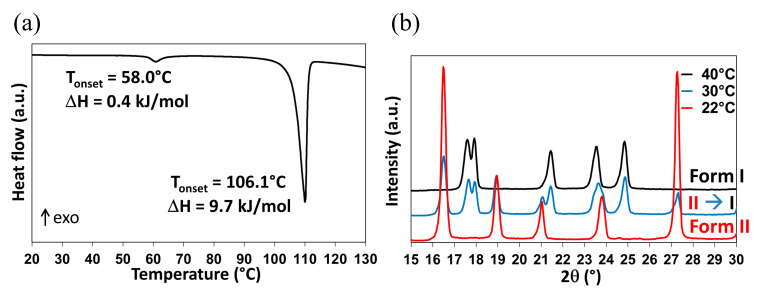
DMU Form II to Form I phase transition observed by (**a**) DSC (the melting peak is also observed at ca. 106.1 °C) and (**b**) TR-XRPD.

**Figure 4 molecules-28-07061-f004:**
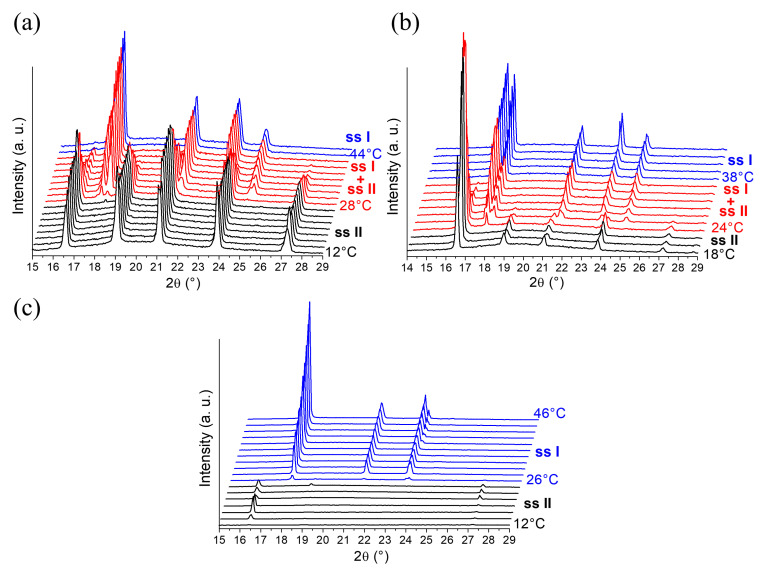
In-situX^®^ diffraction analyses of DMU in suspension in n-heptane with different molar fractions of water: (**a**) 0.15%, (**b**) 0.39%, and (**c**) 12%. The experimental XRPD patterns of pure Form II (ssII), pure Form I (ssI) and the domain of coexistence ssI + ssII are represented in black, blue and red, respectively. For (**a**–**c**), temperature increases from bottom to top in a step of 2 °C from the initial temperature (12 °C or 18 °C) and the first temperature of occurrence of each phase domain is indicated.

**Figure 5 molecules-28-07061-f005:**
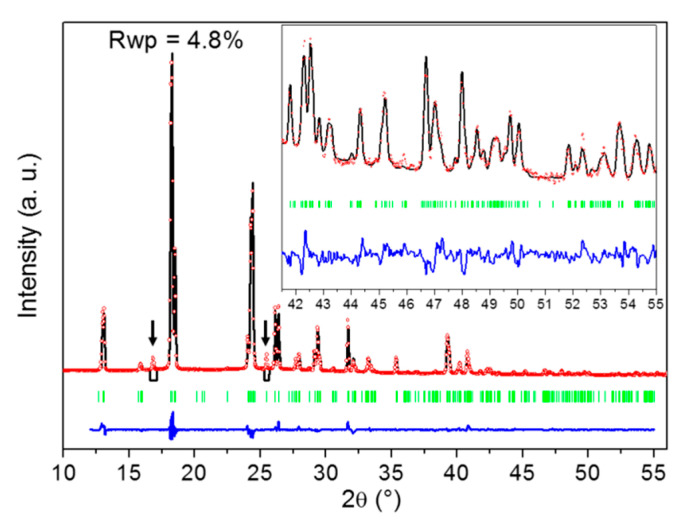
Result of the Rietveld fit of the crystal structure of the DMU monohydrate. The experimental XRPD pattern is in red dots (molar composition in water: 51%; temperature: −20 °C), the calculated one in black, the difference between both in blue and green bars represent the positions of Bragg reflections. The two black arrows point to reflections that do not belong to the XRPD pattern of the DMU monohydrate and that were thus not included in the crystal structure determination procedure. Inset is a zoom (ten times magnified) on the high angle 2θ region [41–55°].

**Figure 6 molecules-28-07061-f006:**
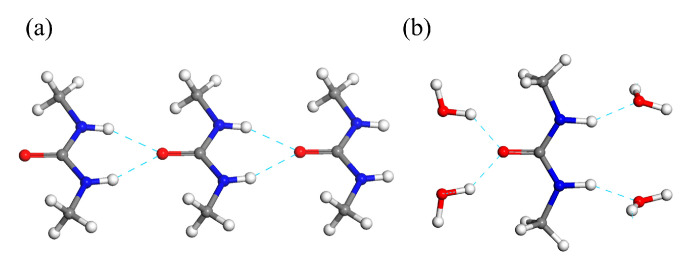
Portion of a molecular chain of (**a**) anhydrous DMU Form II and (**b**) the DMU monohydrate.

**Figure 7 molecules-28-07061-f007:**
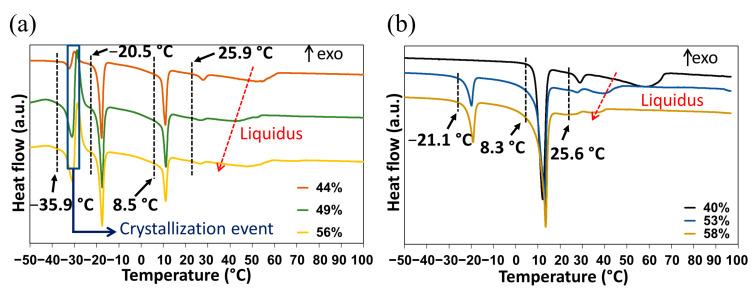
DSC curves of DMU–water mixtures of various compositions in water (expressed in molar fraction of water) (**a**) before and (**b**) after two hours annealing at −10 °C. The onset temperature given for each invariant transition corresponds to the mean value of the onset temperatures measured from the corresponding peaks observed for the mixtures shown.

**Figure 8 molecules-28-07061-f008:**
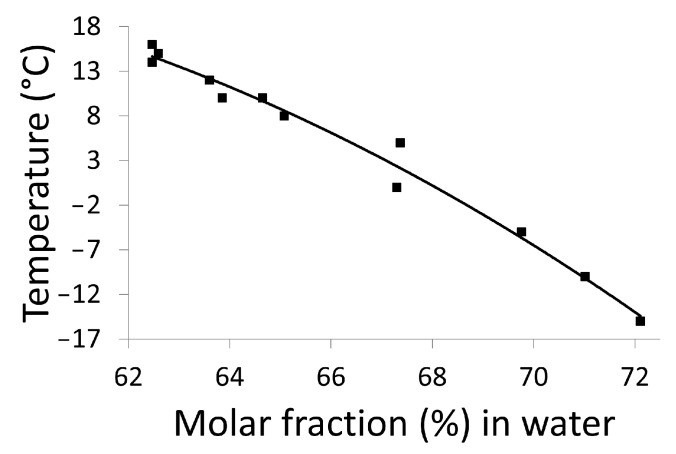
Solubility curve of DMU in water determined by the refractometry method.

**Figure 9 molecules-28-07061-f009:**
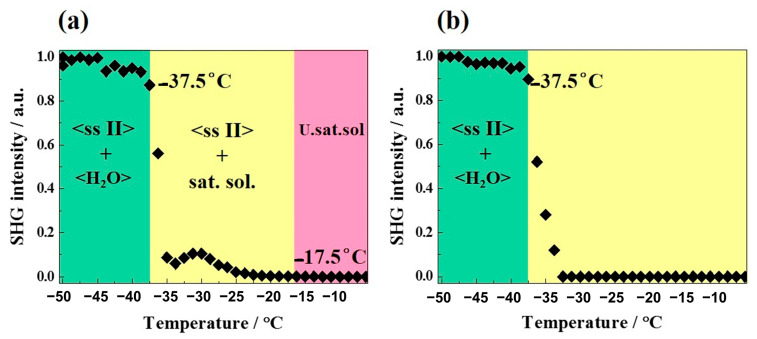
TR-SHG analyses of DMU—H_2_O mixtures of (**a**) 70% and (**b**) 70.5% molar compositions in water. SHG Intensity is normalized to the maximum intensity measured. U.sat.sol. stands for undersaturated solution.

**Figure 10 molecules-28-07061-f010:**
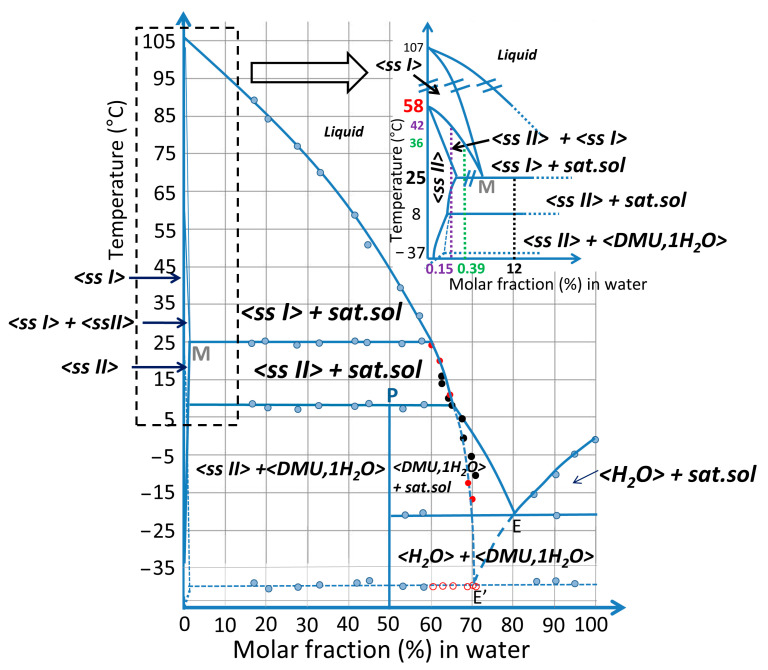
Schematic DMU–water binary phase diagram. Experimental points obtained from DSC (blue dots), refractometry (black dots) and TR-SHG (red dots—open symbols: metastable eutectic temperature; filled symbols: ssII liquidus temperature) analyses are represented. The inset shows a magnification of the DMU-rich side of the phase diagram with the phase domains labelled. In particular, the compositions 0.15%, 0.39% and 12% in molar fraction of water which have been analyzed by In-situX^®^ measurements are highlighted in purple, green and black, respectively. Vertical dashed lines starting from the latter compositions (with the same respective colors) emphasize the phase domains they exhibit as a function of temperature along the ssII to ssI phase transition.

**Table 1 molecules-28-07061-t001:** Crystallographic data determined for the DMU monohydrate together with those of forms I and II of anhydrous DMU.

Form	DMU Monohydrate	DMU Form I [22]	DMU Form II [22]
M (g/mol)	106.13	88.11	88.11
Temperature (°C)	−20	−93	−173
CCDC number	1966048	924553	924554
Crystal system	Monoclinic	Orthorhombic	Orthorhombic
Space group	*C*2/c	*F*dd2	*P*2_1_2_1_2
*a* (Å)	19.1479 (8)	11.3837 (2)	10.8522 (6)
*b* (Å)	13.9557 (5)	19.6293 (4)	4.9102 (3)
*c* (Å)	13.4926 (4)	4.5608 (1)	4.5766 (3)
*β* (°)	134.644 (2)	90	90
Z	16	8	2
Z’	2	0.5	0.5
Volume (Å^3^)	2565 (1)	1019 (1)	244 (1)
Density (g/cm^3^)	1.099 (1)	1.149 (1) 1.120 (1) *	1.200 (1) 1.151 (1) *

* Density (g/cm^3^) calculated from XRPD patterns of anhydrous DMU forms I and II measured at room temperature.

**Table 2 molecules-28-07061-t002:** Hydrogen bond distances and angles between DMU and water molecules for the DMU monohydrate and also between DMU molecules for anhydrous DMU forms I and II. “DMU” and “w” subscripts refer to DMU and water molecules, respectively. Concerning anhydrous DMU forms I and II, O∙∙∙N distances and N–H∙∙∙O angles only involve DMU molecules. The crystal structure of the DMU monohydrate involves two types of molecular chains named type A and type B chains due to the two symmetry independent DMU and water molecules (see Appendix A for a more detailed description of the packing arrangement in the crystal structure of the DMU monohydrate). The hydrogen bond distances and angles between DMU and water molecules belonging to those two molecular chains are reported.

		DMU Monohydrate		DMU Form I [22]	DMU Form II [22]
		Type A Chains	Type B Chains		
O∙∙∙O distances (Å)	O_DMU_∙∙∙O_1w_	2.76 (2)	2.71 (2)		
	O_DMU_∙∙∙O_2w_	2.74 (2)	2.77 (2)		
O∙∙∙N distances (Å)	O_3w_∙∙∙N_1DMU_	2.83 (2)	3.00 (2)	2.85 (2)	2.86 (2)
	O_4w_∙∙∙N_2DMU_	2.88 (2)	2.82 (2)		
N–H∙∙∙O angles (°)	N_1DMU_–H_1DMU_∙∙∙O_3w_	152 (2)	152 (2)	154 (4)	153 (2)
	N_2DMU_–H_2DMU_∙∙∙O_4w_	155 (2)	155 (1)		
O–H∙∙∙O angles (°)	O_1w–_H_1w_∙∙∙O_DMU_	142 (2)	160 (9)		
	O_2w_–H_2w_∙∙∙O_DMU_	138 (4)	163 (4)		

**Table 3 molecules-28-07061-t003:** Characteristics of the different invariants observed by DSC.

T Invariant (°C)	Nature of the Invariant	Stability of the Equilibrium	Phases in Equilibrium
−37	Eutectic	Metastable	<ss II>(<<1%) + <Ice> ↔ doubly saturated liquid (between 70% and 70.5%)
−20	Eutectic	Stable	<monohydrate> + <water> ↔ doubly saturated liquid (≈80%)
8	Peritectic	Stable	<monohydrate> ↔ <ssII> (<<1%) + doubly saturated liquid(≈65%)
25	Metatectic	Stable	<ssII> (<<1%) + doubly saturated liquid (≈60%) ↔ <ss I> (<<1%)

## Data Availability

The data available in this study are available in the present article and Appendix A.

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
