# Peer review of "Critical Influence of Water on the Polymorphism of 1,3-Dimethylurea and Other Heterogeneous Equilibria"

_molecules, 2023, doi:10.3390/molecules28207061_

Round 1

Reviewer 1 Report

Comments and Suggestions for Authors

The manuscript from Baaklini et al. discusses the influence of water in the polymorphic transition of 1,3-dimethylurea (DMU), a building block of biologically relevant molecules. Specifically, DMU describes the solid-to-solid transition from Form II to Form I, which is related by temperature, first reported in current reference 22. The transition temperature is significantly lowered by the presence of water in the system. The observations are supported by a combination of differential scanning calorimetry, temperature-resolved powder X-ray diffraction, in-situX diffraction analyses, refractometry measurements, and second harmonic generation (SHG). The authors also report a DMU hydrate obtained using Rietveld refinement. The results highlight the critical role of water in the thermodynamic transformations of solids, as evidenced by DMU. The results of this work have profound implications for the development of pharmaceutics and electronics based on organic molecules, which often crystallize with water molecules or comparable solvents and are prompt to solid-to-solid transformations. I anticipate the work to be of broad interest to the readership of MDPI Molecules.

Before publication, the authors should address the following gaps in their narrative and methodology:

1) The authors clearly described the dehydration process from the DMU monohydrate into the anhydrous form II. However, the structural transformation from II to I was unclear to this reviewer. The authors should address the suspected molecular motions involved in this solid-to-solid transformation.

2) In Figure 2 in the introduction, the authors show the effect of impurities on polymorphic transitions. It was not immediately clear to me that the “impurity” in the case of DMU is water.  Please make it clear in the text if this is the case.

3) Please provide a diagram for the In-situX diffraction technique to ensure reproducibility from other groups working on similar experiments.

4) Please add the CCDC numbers to Table 1 or the main text for known structures I and II and the monohydrate one.

Reviewer 2 Report

Comments and Suggestions for Authors You have presented a well-founded work. The experimental part has been
carried out rigorously and the interesting results have been presented correctly.

Reviewer 3 Report

Comments and Suggestions for Authors

The manuscript unravels a complicated story about polymorphism and phase diagram of dimethyl urea. The work uses a good combination of various experimental techniques and technically sound. I would recommend publication of the manuscript in Molecules in its current form.

Reviewer 4 Report

Comments and Suggestions for Authors

In the manuscript entitled "Critical influence of water on the polymorphism of 1,3-dimethylurea and other heterogeneous equilibria" the authors raise the important issue of 1,3-dimethylurea polymorphism.  

It is a known fact that 1,3-dimethylurea crystallizes in two enantiotropically-related polymorphic forms (Form I - orthorhombic space group Fdd2 and Form II - orthorhombic P2(1)2(1)2 space group). However, 1,3-dimethylurea has a strong affinity with water and discrepancies on the temperature of transition between polymorphs I and II are in the literature. Considering the above, the authors investigated the polymorphic behaviour of 1,3-dimethylurea with a thorough control of the amount of water in the samples. The DMU–water binary phase diagram was investigated and constructed with the support of several techniques such as DSC, ex-situ and in-situ X-Ray Diffraction and Second Harmonic Generation (SHG). Research carried out by the authors shows that the presence of a small amount of water (hundreds of ppm) led to the large depression of the transition temperature between the two enantiotropically related polymorphic forms of 1,3-dimethylurea (Form II → Form I) from 58 °C down to 25 °C which suggests that thermodynamics rather than kinetics is responsible of this significant temperature drop. Because the authors have observed a singular powder X-ray diffraction pattern that is different from that of the polymorphs, they decided to conduct research on this new solid phase. Due to this the authors crystallized and determined the crystal structure of 1,3-dimethylurea monohydrate determined using powder X-ray diffraction data at sub-ambient temperature. Importantly, the structure of the 1,3-dimethylurea monohydrate has not been previously determined and described in the literature. PXRD measurements show that the 1,3-dimethylurea monohydrate crystallizes in monoclinic C2/c space group. In the crystal structure of this compound, hydrogen bonds between 1,3-dimethylurea and water occur, but there are no hydrogen bonds between 1,3-dimethylurea molecules (observed in the crystals of polymorphs I and II of 1,3-dimethylurea). In summary, the research described in the article shows that the study of heterogeneous equilibria, especially the binary phase diagram, is important for understanding the polymorphic behaviour of crystalline materials. 

Manuscript is very well written, and the results are clearly presented. The conclusions are consistent with the evidence presented and the references are appropriate. 

In my opinion article will be interesting for readers of Molecules journal.  

In conclusion, I recommend the presented manuscript for publication in present form.
